# Molecular Characterization of Hepatitis B Virus in People Living with HIV in Rural and Peri-Urban Communities in Botswana

**DOI:** 10.3390/biomedicines12071561

**Published:** 2024-07-14

**Authors:** Bonolo B. Phinius, Wonderful T. Choga, Motswedi Anderson, Margaret Mokomane, Irene Gobe, Tsholofelo Ratsoma, Basetsana Phakedi, Gorata Mpebe, Lynnette Bhebhe, Tendani Gaolathe, Mosepele Mosepele, Joseph Makhema, Roger Shapiro, Shahin Lockman, Rosemary Musonda, Sikhulile Moyo, Simani Gaseitsiwe

**Affiliations:** 1Botswana Harvard Health Partnership, Gaborone Private Bag BO320, Botswana; bphinius@mail.bhp.org.bw (B.B.P.); smoyo@bhp.org.bw (S.M.); 2School of Allied Health Professions, Faculty of Health Sciences, University of Botswana, Gaborone Private Bag UB0022, Botswana; mokomanem@ub.ac.bw (M.M.); gobei@ub.ac.bw (I.G.); 3Africa Health Research Institute (AHRI), Private Bag X7, Congella, Durban 4013, South Africa; 4The Francis Crick Institute, 1 Midland Road, London NW1 1AT, UK; 5Faculty of Medicine, University of Botswana, Gaborone Private Bag UB0022, Botswana; 6Department of Immunology and Infectious Diseases, Harvard T. H. Chan School of Public Health, Boston, MA 02115, USA; 7Division of Medical Virology, Faculty of Medicine and Health Sciences, Stellenbosch University, Private Bag X1, Matieland, Cape Town 7602, South Africa; 8School of Health Systems and Public Health, University of Pretoria, Private Bag X20, Pretoria 0028, South Africa

**Keywords:** hepatitis B virus, occult hepatitis B, genotypes, escape mutations, Botswana, Africa

## Abstract

(1) Background: Hepatitis B virus (HBV) sequencing data are important for monitoring HBV evolution. We aimed to molecularly characterize HBV sequences from participants with HBV surface antigen-positive (HBsAg+) serology and occult hepatitis B infection (OBI+). (2) Methods: We utilized archived plasma samples from people living with human immunodeficiency virus (PLWH) in Botswana. HBV DNA was sequenced, genotyped and analyzed for mutations. We compared mutations from study sequences to those from previously generated HBV sequences in Botswana. The impact of OBI-associated mutations on protein function was assessed using the Protein Variation Effect Analyzer. (3) Results: Sequencing success was higher in HBsAg+ than in OBI+ samples [86/128 (67.2%) vs. 21/71 (29.2%)]. Overall, 93.5% (100/107) of sequences were genotype A1, 2.8% (3/107) were D3 and 3.7% (4/107) were E. We identified 13 escape mutations in 18/90 (20%) sequences with HBsAg coverage, with K122R having the highest frequency. The mutational profile of current sequences differed from previous Botswana HBV sequences, suggesting possible mutational changes over time. Mutations deemed to have an impact on protein function were _tp_Q6H, _surface_V194A and _preC_W28L. (4) Conclusions: We characterized HBV sequences from PLWH in Botswana. Escape mutations were prevalent and were not associated with OBI. Longitudinal HBV studies are needed to investigate HBV natural evolution.

## 1. Introduction

Hepatitis B virus (HBV) infection remains a significant global health concern particularly in Africa, which has the highest HBV prevalence globally at 5.8% [1]. In Botswana, the prevalence of hepatitis B surface antigen (HBsAg) varies by region, reaching levels as high as 22% in certain areas of the country [2]. Only three studies have reported occult hepatitis B infection (OBI) prevalence in Botswana, with a range from 6.6% to 33% in adults [2,3,4]. OBI is defined as the presence of replicative competent HBV deoxyribonucleic acid (DNA) in the blood and/or liver of individuals testing negative for HBsAg [5].

Understanding the genetic diversity of HBV within specific geographic locations is important for devising effective prevention and control strategies as HBV genotypes have clinical relevance. HBV genotypes are associated with vaccine efficacy [6], treatment response [7], tendency of chronicity [8], HBsAg and hepatitis B e antigen (HBeAg) seroconversion [9]. Previous studies have identified HBV subgenotypes A1, D2 and D3 and genotype E in Botswana [3,10,11]. Notably, the prevalence of subgenotype A1 varies among different demographic groups [3,10,11]. There is need for HBV surveillance and molecular characterization efforts, especially in settings with high human immunodeficiency virus (HIV) prevalence and widespread antiretroviral treatment (ART) use, in which HBV drug resistance can be common [12]. Furthermore, HBV genomics are important in studying vaccine and treatment response, as well as transmission dynamics within the country.

OBI is not reported in national, regional and global reports; however, it has clinical relevance. OBI was determined to be an independent risk factor for hepatocellular carcinoma (HCC) in one study [13]. Drug resistance associated mutations have been identified in participants with OBI [12]. Furthermore, infants born to mothers with positive HBsAg (HBsAg+) were diagnosed with OBI in one study [14]. One mechanism postulated to lead to the OBI phenotype is the presence of mutations that impair HBsAg detection. Some OBI-associated mutations in different HBV open reading frames (ORFs), (preS1, preS2, surface, core, pre-core, X, and the polymerase domains) have been identified and studied in Botswana and South Africa [15,16]. The impact of these mutations on protein function was assessed using available online in silico tools as screening methods for potential candidates of functional in vitro studies. The Protein Variation Effect Analyzer (PROVEAN) was more accurate than other tools being studied [15,16].

In Botswana, the HBsAg positivity remains high, and the OBI prevalence is three to four times that of HBsAg positivity, as prior studies have reported an adult OBI prevalence of 6.6% to 33% [2,3,4]. Therefore, the genetic diversity of HBV in both HBsAg+ and OBI participants needs to be further studied. We aimed to molecularly characterize HBV sequences from people who tested positive for HBsAg and OBI, and to determine the impact of OBI-associated mutations on protein function in a cohort of people living with HIV (PLWH) in Botswana.

## 2. Materials and Methods

### 2.1. Study Population

Archived plasma samples from participants in the Botswana Combination Prevention Project (BCPP) that had previously tested positive for HBsAg (HBsAg+) and OBI (OBI+) were used [2]. Details of the BCPP study are described elsewhere [17]. Briefly, this BCPP study was a pair-matched cluster-randomized study that enrolled 12,610 consenting adults residing in a random sample of ~20% of households in 30 geographically dispersed villages throughout Botswana between the years 2013 and 2018. The main aim of the BCPP study was to assess if a combination of HIV prevention strategies would reduce HIV incidence at a community level compared to the standard of care. At baseline, 3596 BCPP study participants were PLWH and 83% of them knew their HIV status [17]. Our study was approved by the Human Research Development Committee (HRDC) at the Botswana Ministry of Health (HRDC number: 01028).

### 2.2. Laboratory Procedures

HBV screening is described in our previous report [2]. Briefly, available plasma samples from 3304/3596 (91.9%) of PLWH in the BCPP cohort were screened for HBsAg and total core antibodies (anti-HBc). HBsAg+ samples were further screened for HBeAg and immunoglobulin M core antibodies (anti-HBc IgM). HBV viral load was quantified in HBsAg+ samples with sufficient sample volume and performed in samples that tested negative for HBsAg to determine OBI prevalence using the Roche COBAS Ampliprep/Taqman Analyzer (Roche Diagnostics, Mannheim, Germany) [2].

The QIAamp DNA Blood Mini kit (Qiagen, Hilden, Germany) was used to extract DNA from 200 μL of HBsAg+ and OBI+ plasma samples according to the manufacturer’s protocol with a final elution volume of 30 μL. HBV DNA was amplified using tiling primers adopted from Choga’s protocol [18] (Appendix A). Briefly, two 10 μM pools of tiling primers were prepared. A master mix for each primer pool with 5 μL of DNA template was prepared, and HBV DNA was amplified using a protocol from our previous report [12]. After amplification, these PCR products were combined, and library preparation followed [12,18,19]. The library was loaded into flow cells version R9.4.1 (Oxford Nanopore Technologies, Oxford, UK) and the GridION platform (Oxford Nanopore Technologies, Oxford, UK) was used for sequencing.

### 2.3. Sequence Analyses

#### 2.3.1. Genotypic and Mutational Analysis

Generated FASTQ files were uploaded into Genome Detective version 2.64 for reference-based assembly of HBV [20] (last accessed 20 April 2023). The generated consensus sequences were downloaded, and they were viewed, trimmed and aligned in AliView version 1.26 [21]. Geno2pheno version 2.0 (https://hbv.geno2pheno.org) (last accessed 11 December 2023), was used to assign HBV genotypes/subgenotypes. Genotypes were confirmed using phylogenetic analysis. BCPP-generated sequences and reference HBV sequences from GenBank were used to construct a phylogenetic tree of the complete surface ORF using Bayesian Markov chain Monte Carlo (MCMC) in the Bayesian Evolutionary Analysis by Sampling Trees (BEAST) version 1.8.2 with a chain length of 100,000,000 and sampling every 10,000 generations. The analysis utilized an uncorrelated log-normal relaxed molecular clock, the Hasegawa, Kishino and Yano (HKY) model, the general time-reversible model with gamma-distributed rates of variation among sites, and a proportion of invariable sites (GTR+G+I). Tracer v1.7 (BEAST Developers) was used to visualize results and confirm chain convergence. Tree Annotator v1.7.3 (BEAST Developers) was used to choose the maximum clade credibility tree after a 10% burn-in. The mutational profile of sequences generated form this study was compared to mutations in sequences generated in previous Botswana studies that were predominantly from PLWH (2009–2015) [3,10,11,22]. The accession numbers for Botswana references sequences used in this study are KR139680–KR139749, MF979142–MF979176, MH464807–MH464856, MG977689, MG977690 and MG977693–MG977701. The accession numbers for other reference sequences are AY233282, AY576433, FM199980, FM199981, FM200180, FM200189, FM200214, FN821500, JX144294, KF476020, KF849717, KF849730, KM375052, KM375057, KM375058, KM375063, KM375070, KM375072, KM375138, KM375144, KM375168, KM375169, KM375318, KM375718, KM391914, KM519452, KX648547, KX648548, KX982113, KX982130, KX982142, MH347485, MH607866, MK127847, MK127857, MN080536, MN651979 and MW322670.

#### 2.3.2. Impact of Occult-Associated Mutations on Protein Function

Sequences with a depth of >100 were used for this analysis. The Protein Variation Effect Analyzer (PROVEAN), available at http://provean.jcvi.org/index.php (accessed 24 April 2024), was used to determine the impact of occult-associated mutations on protein function. Occult-associated mutations were defined as mutations identified only in OBI+ sequences and those that were overrepresented in OBI+ sequences versus HBsAg+ sequences.

## 3. Results

### 3.1. Participants Clinical Characteristics

Table 1 summarizes the clinical characteristics of participants whose plasma samples were used in the analysis. Most participants were female (66.4%) and had a median age of 43 (IQR: 36–49). Most participants had a low HBV viral load of <2000 IU/mL (58.9%). Approximately 94.4% of participants were on ART and were mostly on a tenofovir disoproxil fumarate (TDF)-containing regimen (61.6%). The TDF regimen also had emtricitabine (FTC) for all participants except for one who was on a dolutegravir/TDF regimen. The majority of participants had undetectable HIV-1 RNA (<40 copies/mL) (88.8%). Median duration time on ART was 7 years (IQR: 4.7–9.9).

### 3.2. Sequencing Success

Figure 1 shows the total number of sequences generated. Sequencing was attempted on 128 samples out of the 271 HBsAg+ samples and the success rate was 67.2% (86/128). Success rate for OBI+ was 29.2% (21/72). Samples that were successfully sequenced had HBV viral loads ranging from target not detected (TND) to >1.7 × 10^8^ IU/mL.

In total, 27 out of the 30 BCPP study sites contributed HBV sequences to this analysis (Appendix A). Samples from participants residing in Mmadinare, Molapowabojang and Otse (BCPP study sites) were not successfully sequenced. Most sequences were generated from participants in the Central district with 50 sequences, followed by the North-West district with 26 sequences. Kgatleng district had the least number of sequences generated (n = 5) (Figure 2).

### 3.3. Genotypic Analysis

Overall, 93.5% (100/107) sequences were genotype/subgenotype A1, 2.8% (3/107) were D3 and 3.7% (4/107) were E. Among the HBsAg+ sequences, 93.0% (80/86) sequences were genotype/subgenotype A1, 3.5% (3/80) were D3 and 3.5% (3/80) were E. For OBI sequences, 95.2% (20/21) sequences were genotype A1 and 4.8% (1/21) were E. Sequences generated in this study clustered randomly by district, HBV viral load and treatment status (Figure 3). All previous HBV sequences from Botswana clustered together and BCPP sequences clustered randomly (Appendix A).

### 3.4. Mutational Analysis

#### 3.4.1. Escape Mutations

A total of 13 escape mutations were detected in 20% (18/90) of sequences with surface coverage (Table 2). _surface_K122R had the highest frequency in 9/18 (50%) of participants with escape mutations. _surface_T114S, _surface_S114L, _surface_C139R, _surface_N146S and _surface_C147Y are associated with impaired virion secretion. Vaccine escape mutations (_surface_G130C, _surface_N131T and _surface_T121N) were identified in four participants. We also identified mutations that may impact HBsAg detection (_surface_T118M, _surface_C121R, _surface_K122R and _surface_G130C) (Table 2). There was no noticeable trend in clinical characteristics of participants with escape mutations. A description of these participants is provided in Appendix A.

#### 3.4.2. Comparison of Botswana Reference HBV Sequences and BCPP HBV Sequences

For all downstream analyses, only subgenotype A1 sequences were used, as they constituted >93% of the sequences. Table 3 and Table 4 show previously generated Botswana sequences (Reference-unique) and BCPP unique mutations with a prevalence of >20%. The full list of mutations is shown in Appendix A. BCPP sequences tended to have more unique mutations in all ORFs with a high prevalence (>20%) that were not observed in previous Botswana sequences. Some amino acid substitutions unique to BCPP sequences had a much higher prevalence, such as _preC_V17F (53.6%), _x_P33S (42.2%) and _surface_I195M (55.7%). Among mutations that were identified in both sets of sequences, the prevalence of _pres2_A7T and _pres2_A11T was higher in the BCPP sequences (19.9% and 10.3% vs. 5.3% for both mutations in the reference sequences). However, _pres2_T38I was lower in the BCPP sequences (30.3% vs. 41%). For the transcriptional transactivator protein (HBx), the prevalence of _x_G22S, _x_A21T and _x_S46P was higher in the BCPP sequences than in the reference sequences (43.8%, 28.1% and 67.2% vs. 8.3%, 8.3% and 41.7%). In the surface protein, _surface_N131T and _surface_V194A had a lower prevalence in the BCPP sequences compared to the reference sequences (2.2% and 7.7% vs. 37.5% and 14.8%), while the opposite was observed for _surface_K122R (10.1% vs. 2.3%).

In Table 4, we report mutations specific to the polymerase domains. It is noticeable that resistance-associated mutations (_rt_M204V, _rt_L180M and _rt_V173L) were unique to the BCPP cohort. In the same RT region, some mutations had a noticeably higher prevalence in the reference sequences as compared to the BCPP sequences. These are _rt_V7A (40.0% vs. 16.3%), _rt_L53I (35.0% vs. 11.4%), _rt_H122N (35.0% vs. 2.4%), _rt_N332S (39.3% vs. 15.9%) and _rt_Q333K (44.4% vs. 14.5%). In the terminal protein (TP) region, _tp_H182Q had a noticeably higher prevalence in the BCPP cohort than in the reference sequences (21.7% vs. 5.1%). _sp_Y86H, _sp_S125N and _sp_S129N in the spacer domain occurred more frequently in the BCPP sequences than in the reference sequences. In the RNase H domain, _RNaseH_Y116F was observed at a much higher prevalence in the BCPP sequences compared to the reference sequences (30.0% vs. 8.3%).

#### 3.4.3. Impact of Occult-Associated Mutations on Protein Function

For this analysis, we focused on sequences that had a depth of >100 and we identified mutations that were in sequences isolated from OBI participants only (_core_T142S, _tp_E88R, _tp_Q6H, _rt_M250L and _preC_W28L). Other mutations were overly represented in OBI+ sequences compared to HBsAg+ sequences. _surface_V194A and _surface_S55P appeared in 3/13 (23.1%) OBI participants each versus 1/53 (1.9%) HBsAg participants each, *p*-value 0.004. Using PROVEAN, three mutations were deemed deleterious, that is, they were deemed to affect protein function negatively: _tp_Q6H, _surface_V194A and _preC_W28L as shown in Table 5.

## 4. Discussion

In this study, we identified HBV subgenotypes A1, D3 and E across a wide geographic area in Botswana, with subgenotype A1 representing more than 93% of all sequences. We also report the mutational profile of HBV in the Botswana population including mutations with deleterious impact on protein function in participants with OBI.

Our findings are consistent with prior HBV studies in Botswana that identified the same subgenotypes, however with varying genotype prevalence [3,10,11]. We also report immune, vaccine, and diagnostic escape mutations in this population, some of which have been identified in other populations in Botswana [10,11,22]. _surface_K122R was the most prevalent escape mutation and was identified only among participants with HBsAg-positive serology. This mutation is associated with decreased HBsAg expression and HBsAg detection failure [27,28,29]; however, it was not detected among the OBI samples. We did not perform quantitative HBsAg ELISA, which could have revealed HBsAg levels in samples with these mutations compared to those without. We also identified known mutations (_surface_T118M and _surface_N146S), and uncharacterized mutations at positions associated with immune escape (_surface_C121R, _surface_Q129C and _surface_G130C) in OBI participants although these mutations are not unique to individuals with OBI in other studies [24,35,39,40]. All these mutations were identified in the major hydrophilic region (MHR) (position 99 to 169) of the HBsAg, with the majority of these being found in the ‘a’ determinant of the MHR (position 124 to 147), which is a major cluster of antigenic epitopes [41]. We also identified vaccine escape mutations (VEMs) in 16.7% of participants with escape mutations, which is a cause for concern, as these may counteract vaccination efforts in the country. This was at position 131 of the surface region, also in the ‘a’ determinant of the MHR, known for mutations that allow for the virus to evade vaccine-induced immune response [42]. We identified a potential vaccine escape mutation, G130C, which was reported as a novel mutation in 2017 [43]. Mutations at position 130 are reported to be VEMs and have been isolated in vaccinated individuals [32,33].

There was a change in mutation patterns between sequences previously generated in Botswana (2009–2015) [3,10,11,22] and sequences we generated in the BCPP study (2013–2018). For example, the RT region of BCPP sequences harbors more drug resistance-associated mutations than the reference sequences. Over 90% of PLWH among the BCPP participants were ART-experienced, while in the previous studies, participants were mostly ART-naïve [3,10,11,22]. Due to the overlapping pattern of the HBV genome, some of the mutations observed in the RT region of the polymerase affected the surface region. Position _surface_I195 corresponds to the _rt_M204 [44]; therefore, its prevalence is higher in BCPP sequences compared to previous Botswana sequences. Furthermore, _surface_E164D is known to alter HBsAg antigenicity and tends to occur with _surface_I195M, as observed in our study and a previous study [45].

Most of the mutations identified in our study are uncharacterized. Other mutational variations of interest between BCPP and previously generated Botswana sequences are the _x_P33S in the X region. This mutation was only observed in the BCPP sequences and had not been previously identified in Botswana. It is a B-cell epitope mutation that has been shown to result in increased endoplasmic reticulum (ER) stress [46] and reduced protein stability in combination with other mutations [47]. There were some mutations that were common to both sequence datasets; however, they were more prevalent in the BCPP sequences—for example, the _x_G22S, which is reported to be an HCC-related HBx mutation [48]. BCPP sequences also had the _x_T36A, a functionally characterized HCC-associated mutation at >20% prevalence. This mutation is reported to enhance viral genome integration into the host cell, resulting in insertion mutations and a 3′-terminal truncation of HBx [49,50]. While previously generated Botswana sequences and the BCPP sequences were generated from different parts of the country, we cannot rule out the possibility that more people are progressing to chronicity and HCC in the population.

To further elucidate on OBI-associated mutations, we used a freely available online tool, PROVEAN, previously shown to be more accurate in predicting the impact of functionally characterized HBV mutations on protein function than other prediction tools [15,16]. This analysis also allows for the selection of mutations that could be candidates for further in vitro studies. Three OBI-associated mutations were deemed deleterious: _tp_Q6H, _surface_V194A and _preC_W28L. The _tp_Q6H mutation has not been characterized; however, it falls within the N-terminal helices of the TP. Mutations and deletions in this subdomain were shown to impact RNA packaging, DNA synthesis and protein priming [51]. _preC_W28L has not been characterized; however, a stop codon at this position has been identified in a participant with OBI [52] and in patients with HCC in a much older study [53]. Other surface mutations (_surface_S55P and _surface_V194A) were overrepresented in OBI+ sequences compared to HBsAg+ sequences, and other studies also show that these mutations are not unique to OBI+ sequences [24,54]. It is worth exploring the HBsAg levels in HBsAg+ and OBI+ samples with these mutations. _surface_S55P has not been functionally characterized, while _surface_V194A is a well-known mutation associated with decreased extracellular HBsAg levels [55].

Our analysis focused on the differences between sequences from HBsAg+ and OBI+ infection, but it could be that the host factors have a key role to play in the OBI phenotype. Host factors such as immune response, human leukocyte antigen classA2 and interleukin-10 have been associated with OBI persistence [25,56,57]. Some studies attribute this phenotype to viral factors such as epigenetic control mechanisms [58,59] and mutations [15,16,60] that lead to reduced HBsAg expression and DNA replication.

A strength of this study is that we used samples from the BCPP study, which recruited participants from different communities in Botswana. However, we analyzed sequences from only 27 out of the original 30 BCPP study sites due to insufficient volumes and the low sequencing success rate. Sequencing success was generally low, especially for OBI+ samples; however, this study still provides the largest number of HBV sequences in Botswana. We generated HBV sequences from participants with a target-not-detectable viral load result, a group usually excluded during sequencing. Escape mutations _surface_T114S, _surface_N131T and _surface_K122R were identified in these participants, which shows that there may be an underrepresentation of mutations in different reports that may not sequence participants with these viral load results. There is need, however, to adopt a nucleic acid testing assay with a lower limit of detection than the one used in our current study. A limitation of this study is that participants were all PLWH predominantly on ART, which limits generalizability to the Botswana population. We made comparisons between previously generated Botswana sequences and the BCPP sequences, which comes with a limitation, as these datasets are not from the same communities. Therefore, we cannot rule out the impact of host genetic factors, which could limit the conclusions in the differences we note. However, the mutational profiles of these sequences are fully documented in this study. HBV vaccination records were not collected in the parent BCPP study; therefore, we cannot ascertain the vaccination status of participants. However, with the universal infant HBV vaccination being introduced in Botswana in the year 2000 and with the youngest participant included in this analysis being 22 years old at the time of BCPP enrollment (2013–2018), it is unlikely that any of the participants were vaccinated.

## 5. Conclusions

We molecularly characterized HBV from PLWH in Botswana and identified that subgenotype A1 is predominant countrywide. We also reported VEMs, which shows the importance of periodic monitoring of circulating HBV strains in the population. It is essential to generate HBV sequencing data to monitor the evolution of HBV and the emergence of mutations that could evade immunity and vaccines, potentially affecting HBV prevention and management strategies.

## Figures and Tables

**Figure 1 biomedicines-12-01561-f001:**
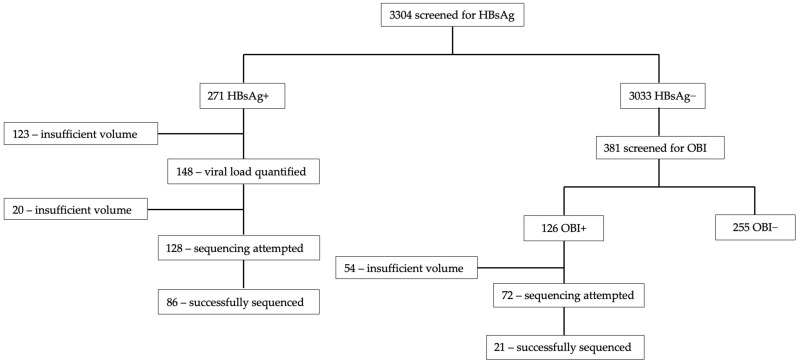
Consort diagram of successfully sequenced samples.

**Figure 2 biomedicines-12-01561-f002:**
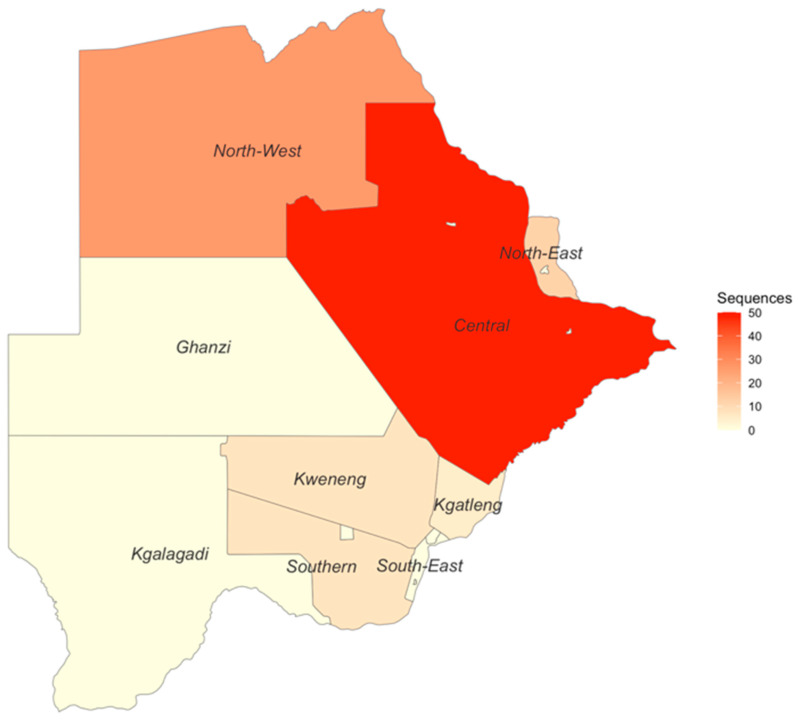
Generated sequences in the different districts of Botswana.

**Figure 3 biomedicines-12-01561-f003:**
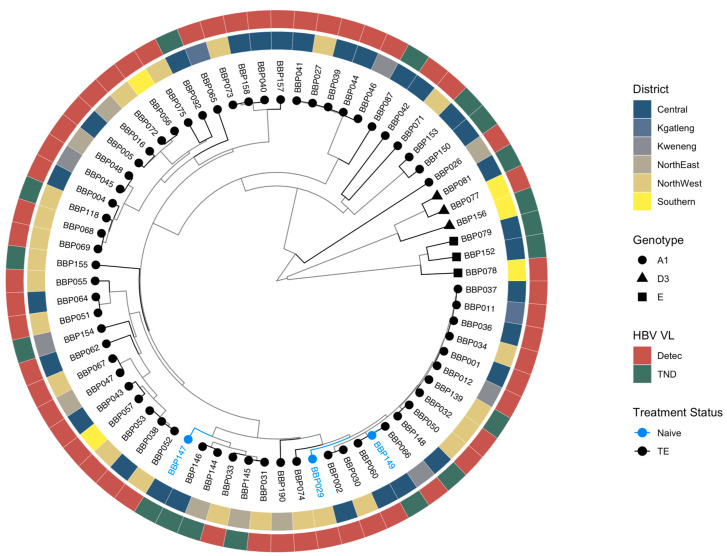
Bayesian phylogenetic tree of sequences generated in the BCPP cohort. HBV: hepatitis B virus; VL: viral load; Detec: detectable; TND: target not detected; TE: treatment experienced.

**Table 1 biomedicines-12-01561-t001:** Participants clinical and socioeconomic characteristics at enrollment.

Characteristics	Number (%) *n* = 107
Sex Female	71 (66.4)
Age, years; median (IQR)	43 (36–49)
HBV type	
HBsAg+	86 (80.4)
OBI+	21 (19.6)
Total anti-HBc, *n* = 104 Positive	87 (83.7)
Anti-HBc IgM status *, *n* = 81 Positive	5 (6.2)
HBeAg status *, *n* = 82 Positive	15 (18.3)
HBV viral load	
Target not detected	21 (19.6)
<2000	63 (58.9)
≥2000	23 (21.5)
HIV viral load	
Undetectable	95 (88.8)
Detectable	12 (11.2)
ART status	
Naïve	6 (5.6)
On ART	101 (94.4)
ART regimen	
No 3TC/TDF-containing regimen	1 (1.0)
3TC-containing regimen	27 (26.7)
TDF-containing regimen ^#^	45 (44.6)
Unknown	28 (27.7)
Duration on ART, years, *n* = 76; median (IQR)	7.0 (4.7–9.9)

HBV: hepatitis B virus; IQR: interquartile range; HBsAg: hepatitis B surface antigen; OBI: occult hepatitis infection; HBeAg: hepatitis B e antigen; anti-HBc: hepatitis B core antibody; anti-HBc IgM: hepatitis B core antigen immunoglobulin M antibodies; HIV: human immunodeficiency virus; ART: antiretroviral therapy; 3TC: lamivudine; TDF: tenofovir disoproxil fumarate. * Only in participant with HBsAg-positive serology. ^#^ The TDF-containing regimen also had emtricitabine (FTC) for all participants except one who was on a dolutegravir/TDF regimen.

**Table 2 biomedicines-12-01561-t002:** Escape mutations in the surface region.

Mutation	Frequency	Genotype	HBV Type	Reported Impact	References
T114S	2	A1	HBsAg	Other substitutions at position 114 (R) reported to impair virion secretion	[23]
S114L	1	E	HBsAg	Other substitutions at position 114 (R) reported to impair virion secretion	[23]
T118M	1	A1	OBI	Impair antigenicity, detection escape	[24]
C121R	1	A1	OBI	Other substitutions at position 121 (S) reported to reduce antigenicity and impair HBsAg detection	[25,26]
K122R	9	A1	HBsAg	Decreased HBsAg expression, detection failure	[27,28,29]
Q129C	1	A1	OBI	Other Q129 (N) mutations lead to impaired antigenicity and immunogenicity, Q129R leads to impaired virion/S protein secretion, Q129H leads to decreased virion secretion	[23,30,31]
G130C	1	A1	OBI	Other G130 mutations to lead to diagnostic escape, vaccine/ immunoglobulin therapy escape, altered antigenicity	[32,33]
N131T	2	A1	HBsAg	Vaccine escape	[34]
T131N	1	D3	HBsAg	Vaccine escape, diagnostic escape, hepatitis B immunoglobulin resistance	[35,36,37]
C137I	1	A1	HBsAg	Other C137 mutations are reported to decrease antigenicity	[25]
C139R	1	A1	HBsAg	Impair virion/S protein secretion	[31]
N146S	1	E	OBI	Impair virion secretion	[23,38]
C147Y	1	A1	HBsAg	Impair virion secretion	[23]

HBsAg: hepatitis B surface antigen; OBI: occult hepatitis infection.

**Table 3 biomedicines-12-01561-t003:** Mutations in HBV ORFs.

	Reference Sequences			BCPP Sequences			Common Mutations		
	Mutation	Frequency	Prevalence	Mutation	Frequency	Prevalence	Mutation	Reference	BCPP
PreS1	A90P	9/37	24.3%	I48V	21/99	21.2%	T94P	44.4%	50.5%
				A90V	23/99	23.2%			
PreS2	None			I45T	18/89	20.2%	A7T	5.3%	16.9%
				L54S	21/89	23.6%	A11T	5.3%	10.3%
							T38I	41.0%	30.3%
							A53V	12.8%	4.5%
preC	None			V17F	15/28	53.6%	None		
Core	None			None			None		
X	V131I	4/19	21.1%	R26C	13/64	20.3%	S11P	12.5%	18.3%
				P29S	17/64	26.6%	G22S	8.3%	43.8%
				P33S	27/64	42.2%	A31T	8.3%	28.1%
				T36A	14/64	21.9%	S46P	41.7%	67.2%
Surface				E164D	25/87	28.7%	K122R	2.3%	10.1%
				I195M	39/70	55.7%	N131T	14.8%	2.2%
							V194A	37.5%	5.7%

BCPP: Botswana Combination Prevention Project

**Table 4 biomedicines-12-01561-t004:** Mutations in the polymerase domains.

	Reference Sequences			BCPP Sequences			Common Mutations		
	Mutation	Frequency	Prevalence	Mutation	Frequency	Prevalence	Mutation	Reference	BCPP
TP				Q138H	15/72	20.8%	V71I	28.2%	34.7%
							Q87H	53.8%	47.2%
							H182Q	5.1%	21.7%
Spacer	Q6K	15/38	39.5%	S18P	12/59	20.3%	P64A	44.7%	36.3%
	A7T	23/38	60.5%	H47R	19/91	20.9%	I84T	13.2%	12.1%
	S89T	9/38	23.7%	P127S	12/52	23.1%	Y86H	13.2%	26.4%
	L158I	10/38	26.3%				H93S	44.7%	45.1%
							S125N	5.6%	22.9%
							S129N	5.3%	12.3%
RT	R110G	12/40	30.0%	N124H	19/84	22.6%	V7A	40.0%	16.3%
	Q139H	12/40	30.0%	Y126H	20/84	23.8%	L53I	35.0%	11.4%
	H271C	13/40	32.5%	V173L	23/84	27.4%	I103V	37.5%	23.5%
				L180M	41/87	47.1%	P109S	20.0%	14.1%
				M204V	37/69	53.6%	H122N	35.0%	2.4%
							W153R	7.5%	19.0%
							K266V	27.5%	14.5%
							K266I	65.0%	65.2%
							N332S	39.3%	15.9%
							Q333K	44.4%	14.5%
RNase H				V128D	18/70	25.7%	S2P	41.7%	61.9%
				V148A	24/70	34.3%	Y116F	8.3%	30.0%
							R151K	8.3%	7.8%

BCPP: Botswana Combination Prevention Project; TP: terminal protein; RT: reverse transcriptase.

**Table 5 biomedicines-12-01561-t005:** Impact of mutations identified only/overrepresented in participants with OBI.

ORF	Mutation	PROVEAN Prediction
Core	_core_T142S	Neutral
Terminal protein	_tp_E88R	Neutral
	_tp_Q6H	Deleterious
Surface	_surface_V194A	Neutral
	_surface_S55P	Deleterious
Reverse transcriptase	_rt_M250L	Neutral
Pre-core	_preC_W28L	Deleterious

ORF: open reading frame; PROVEAN: Protein Variation Effect Analyzer.

## Data Availability

The data generated in this study are available upon request from the corresponding author. The sequences are not publicly available, as they are currently being analyzed for other objectives.

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
