# Peer review of "Molecular Characterization of Hepatitis B Virus in People Living with HIV in Rural and Peri-Urban Communities in Botswana"

_biomedicines, 2024, doi:10.3390/biomedicines12071561_

Round 1

Reviewer 1 Report

Comments and Suggestions for Authors

Dear authors,

The article titled: "Molecular Characterization of Hepatitis B Virus in People Living with HIV in Rural and Peri-Urban Communities in Botswana" is well written and represents data important for local but also with impact on global communities and health. The manuscript has all parts of an original research article; appropriate methodology was used, but some information regarding the seq analysis needs to be included. Accession numbers of reference HBV seq need to be added, and acc. numbers of HBV seq obtained in this study must be added. Fig 3 is small. It would be better if better resolution and a bigger tree in Fig 3 were made. Also, if You have data regarding HBV vaccination in the studied population, it should be added. Other results are ok, and the Discussion is well written in association with the results obtained in this study.

Also, some additional proofreading is needed throughout the manuscript to correct missing letters and words, for example, line 73 in text preS1 and preS2, ...

All the best in Your future work

Comments on the Quality of English Language

Some minor editing is needed (missing letters and similar)

Author Response

Reviewer 1

The article titled: "Molecular Characterization of Hepatitis B Virus in People Living with HIV in Rural and Peri-Urban Communities in Botswana" is well written and represents data important for local but also with impact on global communities and health. The manuscript has all parts of an original research article; appropriate methodology was used, but some information regarding the seq analysis needs to be included.

  1. Accession numbers of reference HBV seq need to be added, and acc. numbers of HBV seq obtained in this study must be added.

Response 1: Accession numbers for references sequences have been included in the methods, Lines 144 – 152. Sequences from this study have not yet been deposited into online databases as they are still being analysed for other objectives of the study. We have included this statement in the Data availability section, Lines 438 – 440. “The data generated in this study are available upon request from the corresponding author. The sequences are not publicly available as they are currently being analyzed for other objectives.”

  1. Fig 3 is small. It would be better if better resolution and a bigger tree in Fig 3 were made.

Response 2: Figure 3 has been replaced with a new figure that shows clustering of HBV sequences generated in the study by genotype, district, treatment status and HBV viral load category. Sequences clustered randomly.The initial Figure 3 is now added to the supplemtary data (Figure S3) and the quality of the figure has been improved.

  1. Also, if You have data regarding HBV vaccination in the studied population, it should be added.

Response 3: We do not have vaccination records of the study participants. This has been added as a limitation in lines 372 – 376 which reads. “HBV vaccination records were not collected in the parent BCPP study therefore we cannot ascertain the vaccination status of participants. However, with the universal HBV infant vaccination being introduced in Botswana in the year 2000 and with the youngest participant included in this analysis being 22 years old at the time of BCPP enrollment (2013-2018), it is unlikely that any of the participants were vaccinated.”

  1. Also, some additional proofreading is needed throughout the manuscript to correct missing letters and words, for example, line 73 in text preS1 and preS2, ...

Response 4: Thank you for the comment, proofreading has been done and corrections have been made.

  1. Comments on the Quality of English Language: Some minor editing is needed (missing letters and similar)

Response 5: The manuscript has been re-checked and corrections have been made where necessary.

Reviewer 2 Report

Comments and Suggestions for Authors

In this study, Phinius et al conduct genomic characterization of hepatitis B virus samples from PLWH in Botswana. They identify sequences from both HBsAg+ and OBI cases, resulting in 107 genomes. The authors find that the majority of sequences from this region were genotype A1, and that a number of the sequences harbored mutations, which may warrant further characterization.

The manuscript is well-written and the wealth of genomes are appreciated for the field. As efforts to develop and deploy HBV countermeasures are furthered, surveillance data such as those presented here are critical. A few additional analyses may be worth looking into and are suggested. The sequencing data needs to be made publicly available before publication.

Major comments:

·       Are all the participants in the study vaccinated against HBV? That would affect our interpretation of the drug resistant mutations that were observed. Have any of the participants experienced HBV symptoms and/or received treatment? Please clarify in the text.

·       Are the genomes that were sequenced correlated with any of the characteristics that are available to the authors? For example, are sequences clustered by district and/or community? Is there clustering between those with detectable viral load versus those undetectable? Between those on ART and not? Between the different ART regimens? Similarly, for these kinds of characteristics, is there a noticeable trend for the mutations that were identified (e.g., specific to a characteristic)? While this would of course be correlative, it may provide a clue into the epidemiology of the virus.

·       The numerous genomes that were collected are very much appreciated. The sequences, and ideally the raw sequencing data from the Nanopore, should be deposited to public repositories, such as GenBank and NCBI SRA, respectively. This would allow their wider usage (and would increase citations to this manuscript!).

Minor comments:

·       Line 32: missing “HIV” after “people living with”

·       Line 115: typo for “suit”

·       Figure S2: There is a box that says “Plot Area” near the middle of the figure

Author Response

Reviewer 2

In this study, Phinius et al conduct genomic characterization of hepatitis B virus samples from PLWH in Botswana. They identify sequences from both HBsAg+ and OBI cases, resulting in 107 genomes. The authors find that the majority of sequences from this region were genotype A1, and that a number of the sequences harbored mutations, which may warrant further characterization.The manuscript is well-written and the wealth of genomes are appreciated for the field. As efforts to develop and deploy HBV countermeasures are furthered, surveillance data such as those presented here are critical. A few additional analyses may be worth looking into and are suggested.

Major comments

  1. The sequencing data needs to be made publicly available before publication.

Response 1: Sequences from this study have not yet been deposited into online databases as they are still being analysed for other objectives of the study. We have included this statement in the Data availability section, Lines 434 – 436. “The data generated in this study are available upon request from the corresponding author. The sequences are not publicly available as they are currently being analyzed for other objectives.”

  1. Are all the participants in the study vaccinated against HBV? That would affect our interpretation of the drug resistant mutations that were observed. Have any of the participants experienced HBV symptoms and/or received treatment? Please clarify in the text.

Response 2: We used samples from the BCPP cohort which was primaily an HIV cohort and HBV symptoms were not captured in this cohort. We have a brief description of BCPP in the methods (Lines 90 – 96) and cited a paper that has the full description (ref 17). We do not have vaccination records of the study participants as the parent BCPP study did not collect the vaccination status of participants. This has been added as a limitation in lines 372 – 376 which reads; “HBV vaccination records were not collected in the parent BCPP study therefore we cannot ascertain the vaccination status of participants. However, with the universal HBV infant vaccination being introduced in Botswana in the year 2000 and with the youngest participant included in this analysis being 22 years old at the time of BCPP enrollment (2013-2018), it is unlikely that any of the participants were vaccinated.” Table 1 shows that 94.4% of participants were on ART, and this point is included in the discussion when discussing the fact that more HBV drug resistance mutations were found in BCPP sequences compared to previous HBV sequences from Botswana,  Lines 308 – 312.

  1. Are the genomes that were sequenced correlated with any of the characteristics that are available to the authors? For example, are sequences clustered by district and/or community? Is there clustering between those with detectable viral load versus those undetectable? Between those on ART and not? Between the different ART regimens? Similarly, for these kinds of characteristics, is there a noticeable trend for the mutations that were identified (e.g., specific to a characteristic)? While this would of course be correlative, it may provide a clue into the epidemiology of the virus.

We provided a description of participants with escape mutations and there is no noticeable trend in terms of participant outcomes, all participants were on ART. This is noted in lines 238 – 239. A supplemental table (Table S3) is provided that describes the charactersitics of participants with escape mutations. We also provide a phylogenetic tree that shows study generated sequences only (Figure 3). There was no noticiable clustering of sequences by district, HBV viral load and treatment status (Figure 3). This has been added to the description of the tree (Lines 207 – 209).

  1. The numerous genomes that were collected are very much appreciated. The sequences, and ideally the raw sequencing data from the Nanopore, should be deposited to public repositories, such as GenBank and NCBI SRA, respectively. This would allow their wider usage (and would increase citations to this manuscript!).

Response 4: Sequences from this study have not yet been deposited into online databases as they are still being analysed for other objectives of the study. We have included this statement in the Data availability section, Lines 434 – 436. “The data generated in this study are available upon request from the corresponding author. The sequences are not publicly available as they are currently being analyzed for other objectives.”

Minor comments

  1. Line 32: missing “HIV” after “people living with”

Response 5: The sentence now reads  “We utilized archived plasma samples people living with human immunodeficiency virus (PLWH) in Botswana” Line 31 – 33.

  1. Line 115: typo for “suit”

Response 6: The sentence now reads “After amplification, these PCR products were combined, and library preparation followed”, Line 120 – 121.

  1. Figure S2: There is a box that says “Plot Area” near the middle of the figure

Response 7: Figure S2 has been replaced